# Effective Communication at Different Phases of COVID-19 Prevention: Roles, Enablers and Barriers

**DOI:** 10.3390/v13061058

**Published:** 2021-06-03

**Authors:** Khayriyyah Mohd Hanafiah, Celine Ng, Abdul Matiin Wan

**Affiliations:** 1Global Health Diagnostics Development, Macfarlane Burnet Institute, Melbourne 3004, Australia; 2School of Biological Sciences, Universiti Sains Malaysia, Minden 11800, Malaysia; celine.ng@student.usm.my; 3Axiom Learning, Mont Kiara 50480, Malaysia; amwan@b2el.com

**Keywords:** communication, science communication, COVID-19, zoonosis, prevention, One Health, public health, pandemic, misinformation

## Abstract

In an age of globalisation and hyperconnectivity, the COVID-19 pandemic has caused unprecedented and sustained impact worldwide. This article discusses issues related to (science) communication at different phases of the COVID-19 epidemic timeline. We consider the role of communication for prevention from the ecological perspective, taking into consideration that many emerging pathogens, including COVID-19, likely arise in part due to anthropogenic changes to natural environments. Communication forms part of the early response setting the scene for public buy-in of public health interventions at the start of an outbreak, as well as to maintain precautions over time. Finally, communication is a key element in increasing acceptance for new tools that require mass uptake to be effective, as seen with roll-out challenges for the COVID-19 vaccines, which faced heightened concerns of efficacy and safety while mired with rampant misinformation. Ultimately, strategies for prevention of viral epidemics such as COVID-19 must include communication strategies at the forefront to reduce the risk of the emergence of new diseases and enhance efforts to control their spread and burden. Despite key themes emerging, what constitutes effective communication strategies for different people and contexts needs to be investigated further.

## 1. Introduction

The coronavirus disease 2019 (COVID-19) pandemic, caused by the highly contagious severe acute respiratory syndrome coronavirus 2 (SARS-CoV-2), rapidly emerged and significantly altered modern life across the globe [1]. The SARS-CoV-2 virus is hypothesised to originate from bats [2], crossing over through several species of wildlife before infecting humans through the angiotensin converter enzyme 2 (ACE-2) receptor [3]. Despite COVID-19 being a new disease, the threat of zoonotic disease transmission is nothing new. Prominent epidemics and pandemics such as swine flu (H1N1), Ebola, SARS, Hendra/Nipah and arguably HIV/AIDS [4], began as zoonotic diseases. Increasing human-wildlife interfaces, arising from habitat loss and fragmentation, tourism activities, hunting, international wildlife trade, and animal research further precipitates emerging threats from future epidemics and pandemics [5,6].

An underlying element of this threat is our limited awareness and appreciation of the anthropogenic factors behind the emergence and persistence of pandemics, which reduces our ability to prevent and manage these risks to public health [7]. Consequently, strategies to mitigate these threats require better communication between different stakeholders, stewards and members of society [8,9]. Often, communication strategies in dealing with outbreaks have been treated as supplementary efforts, rather than being treated as a core component in pandemic preparedness and management. In today’s age of information freedom and social media, argued to be a “fifth” pillar of democracies, there is increasing appreciation that so-called infodemics spread as rapidly, if not faster than and with tantamount impact, as the pandemics they surround [10]. For the first time, due to this collision of the infodemic and pandemic impact of COVID-19, the World Health Organization (WHO) has dedicated significant effort to address risk communication issues under the WHO’s Information Network for Epidemics (EPI-WIN) and the release of a public health research agenda for managing infodemics [10]. While such initiatives to address misinformation represent leaps in the right direction, there remain other aspects of preparedness that also require concerted communication strategies to address different needs at different phases of a pandemic. Thus, this article discusses the role and impact of science communication on COVID-19 pandemic preparedness and prevention, primarily (1) addressing the emergence of zoonotic diseases, (2) pushing for public health action and (3) sustaining public health interventions towards pandemic recovery. Finally, we note significant lessons and remaining questions related to the communications surrounding COVID-19.

## 2. Addressing Emergence of Zoonotic Diseases

### 2.1. Incorporating Public Health in Environmental and Climate Action Communication

While the sustainable development goals (SDGs) were established in 2015 to provide unified interlinked direction for research and development globally, the ideas of “One World, One Health”™, seeking to harmonise disciplines across the life and environmental sciences with a special focus on zoonotic diseases, have been articulated since 2004 if not earlier [11]. In 2013, Jones et al. concluded in a systematic review that “future zoonotic disease emergence or reemergence will be closely linked to the evolution of the agriculture-environment nexus” [12], yet despite 30% higher crop yields, land conversion for agricultural use continues to increase [13]. While more direct impacts of agricultural expansion such as deforestation, biodiversity loss and increase in greenhouse gas emissions are well established [14], its indirect role in altering zoonotic and intermediate host species distribution which leads to viral emergence remains underappreciated [15]. Tellingly, for COVID-19, recent analysis sheds light on increased climate-induced bat species richness in the geographical region considered as the likely origin of the bat-borne ancestors of SARS-CoV-1 and SARS-CoV-2 [16].

Wildlife is more susceptible to contracting or transmitting zoonosis in recently deforested landscapes [17], as the forest fragmentation increases cross-species spillover that may lead to the emergence of zoonotic pathogens. With diminishing ecosystem boundaries between humans and wildlife, interacting species become susceptible to zoonoses through fecal-oral transmission, such as exposure to contaminated soil, food, water, and fomites [18]. Significant land use changes and conversion of primary forest with high biodiversity for agricultural, industrial, and urban development is associated with over 56% increase in documented infectious diseases [19], transmitted directly or through increase in vector species such as rodents [20]. 

High levels of stakeholder awareness and engagement on risks posed by unsustainable development are thus required to ensure effective preventive interventions against future disease outbreaks. While environmental education serves as a valuable form of science communication providing scientific knowledge, building foundations on the importance of biodiversity conservation and highlighting public roles for conservation action [21], key science communication messages in environmental conservation may require reformulation to also acutely address the significance of zoonotic diseases over the past decade including the COVID-19 pandemic, all of which are driven to some part by increased encroachment of wildlife habitats [22]. For years, wildlife biologists and nature educators have been communicating messages for environmental and biodiversity conservation, which traditionally focused on ideas of compassion for other living beings and protecting the planet for future generations through social marketing, awareness and outreach programs, nature-guided tours, volunteerism to engage public participation, and (more recently) virtual engagements [23,24]. While such messages are important and should be maintained, they need to include cautionary tales of more immediate and direct consequences on global human health vis-à-vis the risk of zoonotic diseases. These gaps in public communication may be bridged by teaming up infectious disease epidemiologists and public health experts with social marketing and wildlife and environmental conservation experts, who may co-create more comprehensive awareness and outreach programs. Furthermore, such teams may benefit by further including agricultural engineering and bioeconomy experts for engagement with policy makers, businesses and farmers—arguably the most pivotal groups to communicate the urgency of zoonotic disease prevention through natural habitat management and conservation.

### 2.2. Mitigating Human-Wildlife Interface

In deforested landscapes, wildlife is not only threatened by habitat destruction, poaching and wildlife trade, but also through exposure to human communities where there is overlap between fragmented forests and areas with increasing anthropogenic presence. For COVID-19 in particular, the hypothesised viral spread from bats through pangolins is due to weakening of the intermediate host’s immune system during the wildlife trading process and in captivity, further exacerbated by crowded conditions in wildlife wet markets thereby accelerating interspecies transmission [25]. Thus, on top of creating a collective will to conserve the environment and biodiversity, there is a need to address persisting problematic human–wildlife interactions such as consumption of bushmeat and exotic traditional medicine leading to the establishment of large wildlife wet markets, as well as wildlife feeding and exotic pet trading [26,27,28,29]. 

One opportunity (and challenge) for public engagement centers on modulating perceptions of “normal” public-wildlife contact. In many Southeast Asian countries, macaques living in forest fringes often become reliant on food handouts from the public or tourists who directly feed them believing they are helping care for the animals, and/or through foraging of discarded food scraps that are not appropriately managed [6]. However, direct interaction is not only limited to local areas where deforestation occurs. The global trade for exotic pets, food, clothing, traditional medicine, or laboratory animals only add to the increasing breadth and reach of the human-wildlife interface and the potential for cross-border viral spillovers [28,30]. The establishment of exotic pet cafes, petting zoos and open (illegal) advertisement and glorification of exotic pets on social media exemplify recent trends that only fuel existing drivers of zoonosis [31,32,33]. The high susceptibility of interspecies pathogen transmission between humans and other Hominidae species resulting in a high fatality rate of human-exposed primate populations [34,35], and more recently, mutated SARS-CoV-2 strains infecting a global network of farmed minks (leading to mass culling) [36], further adds to the urgency for awareness and behavior change to mitigate dangers posed by the increasing human-wildlife interface. 

Inadvertently, the profound impacts of COVID-19 outbreak have refocused global attention towards the importance of understanding the origins of the coronavirus and zoonotic transmission risks, and the popularity of consuming bushmeat and exotic medicines in China has declined since the pandemic compared to earlier years [37,38]. Pivoting from this changing public perception, joint effort by conservation advocates, non-governmental organisations (NGOs), wildlife biologists, veterinarians, and local authorities to harmonise regulation and community engagement in order to reduce demand, consumption, and alter social norms involving wildlife may begin to recalibrate the relationships that form the human-wildlife interface, and consequently mitigate future risk of zoonotic disease emergence. 

## 3. Pushing for Public Health Action 

### 3.1. Communicating Risk and Need for Action in the Paucity of Evidence

The COVID-19 pandemic has shown that acknowledging and communicating the problem is the first step to adopting solutions. Barriers to these efforts relate to the challenges in convincing decision-makers pertaining to risks and appropriate mitigation when evidence is absent or limited as was seen early in the pandemic. Further complicating matters is the tangled web of competing and often contradictory communication delivered to the public by various agents. Where the epidemiology of COVID-19 was concerned, governments and public health authorities struggled in the early days of the pandemic to arrive at a consistent narrative that could elucidate risks and mitigating factors surrounding the disease [1,39]. This was particularly evident in the disparity between health advisories issued by agencies under the purview of different governments, some of which were subject to geopolitical antagonism and prejudice.

One of the biggest early controversies that detract from effective action pertains to the airborne nature and effectiveness of mask-wearing for controlling COVID-19 [40]. While person-to-person contact was a known route of transmission, the risks associated with the virus being airborne was not fully appreciated until the first wave of community transmissions across countries were peaking. The conflicting communication on the mode of transmission to some extent was unavoidable due to limited evidence at the time. However, there also appeared to be initial reluctance from authorities and infectious disease experts to confirm airborne risk of COVID-19 and the consequent need for wearing masks, despite early indications that a high percentage of infected individuals were asymptomatic and were from undocumented transmissions [41,42], and that infected people were infectious prior to developing symptoms [43,44]. Indeed, the early calls from the WHO and the United States Centers for Disease Prevention and Control (US CDC) were against mask wearing; this was partly due to worries over rising cost and supply shortage [45] jeopardising PPE supply for healthcare workers, and concerns over reduced public motivation to adhere to social distancing. Additionally, there was underlying resistance to mask-wearing among western nations due to factors such as discomfort and social stigma [46,47]. 

As a result, even after it was evident that aerosolised SARS-CoV-2 persisted in areas with low ventilation [48], and that masks could reduce transmissions particularly in such areas [49], there was delay in adoption, or worse, opposition to adopting mask-wearing especially in Europe and the United States. On top of the need for wearing masks, the manner and materials used for mask-wearing also became a source of contention. However, once evidence showing equivocal benefit of wearing cloth masks were communicated [50,51], the fashion industry and consumers became more empowered to increase supply and uptake of reusable alternatives to surgical masks [52]. In many parts of the world, the practicability, cost-effectiveness, fashionability, and sustainability of reusable cloth masks may have even facilitated uptake of mask-wearing, as seen in the diversity of reusable masks from local designers, small businesses and even those home-made [53,54,55]. It is notable however that even before the effectiveness of masks against COVID-19 were well-evidenced, South Africa, Taiwan, China, Vietnam, Thailand, and many other countries in Asia and Africa mandated public wearing of cloth masks to reduce reliance on surgical and N95 masks [52,56]. In such countries, prior experience with similar infectious disease, air pollution and general public acceptance to mask-wearing were likely the driving factors. Encouragingly, while there remains resistance to mask-wearing among Scandinavian populations [57], the attitudes in other parts of Europe and the United States now matches that in East and Southeast Asian countries. Nevertheless, the use of masks for control of COVID-19 has specifically highlighted the pitfalls of neglecting preliminary information in the absence of solid evidence when communicating science and public health messages related to an emerging and rapidly developing crisis [58].

### 3.2. Ensuring Public Health Communication Results in Meaningful Change in Behavior 

Eventually, as acceptance for mask-wearing along with other precautions such as social distancing increased due to demonstrated and perceived effectiveness [59], there were nuances and gray areas in implementation that required more thoughtful communication. In many settings, non-pharmaceutical interventions were implemented through regulatory enforcement such as strict stay-at-home orders, roadblocks to ban inter-district/state travel, and associated fines for people flouting restriction orders [60,61,62]. While harmonising public health precautions with regulations is an important part of implementation [63], problems arise when regulations are enforced with severe penalties, without strong communication and community engagement to increase public buy-in, science literacy and appreciation of the rationale behind the precautions—particularly where there is low trust in governments [64,65]. 

In particular, lockdown and social distancing rules should not be a wish-list of ideals from governments without considering the experiences of the people and the need to balance practicability with potential and perceived public health gain [66]. Regulations that are vulnerable to variation in interpretation and implementation by enforcement authorities, would otherwise open avenues for abuse of power and excessive infringement of civic freedoms [67]. There have been instances notably in the United States, where the level of regulation associated with COVID-19 control gave rise to public discontent and at times even resulted in protests and demonstrations under crowded conditions [68], which would exacerbate the spread of infectious diseases. While these instances may be more reflective of the worrying disenfranchisement and mistrust that is increasingly pervasive among the public towards public health policies [64,65], better communication and engagement strategies may reduce undue resistance. Further reports of the unequal impact of mobility restrictions on people who are able to work from home compared to those who relied on daily wages, the disparity in penalties associated with flouting COVID-19 related orders which reflected biased treatment towards the privileged, for e.g., the notorious “Cummings incident” in the UK [69], and economic vulnerability of low-income earners in the community together send negative messages that precipitate discontent and resistance against otherwise potentially effective measures [69,70].

In fact, previous work informed by the health belief model [71] and protection motivation theory [72] has shown that people will only act on health warnings if they: (1) believe that they are personally susceptible to develop the condition against which protection is required; (2) perceive the condition as severe; (3) perceive the preventive action as effective to reduce the threat; and (4) believe they are capable to perform the preventive action [73]. However, when applying this insight at a grander scale as seen in efforts to curtail COVID-19, a “one size fits all” approach in the form of “do’s and don’ts” proves unrealistic and counterproductive given the dynamic scenarios characterising heterogeneous populations within open real-world systems. Likewise, bombardment of the public with information, falsely assumed to be an adequate trigger for meaningful action (“knowing is half the battle”), contributes instead to greater anxiety, cognitive avoidance strategies and defensive biases relating to health-based judgments [74]. This is especially relevant when each community member’s social identity needs in interaction with other contextual factors e.g., socioeconomic or cultural pressure can increase and mitigate actual rejection of evidence—or so-called knowledge resistance [75]. 

A more effective strategy may be to communicate the concept of individual COVID-19 risk and harm reduction based on dynamic social situations rather than ideals [75], while working towards building public trust in authorities [76,77]. Community engagement with locale-specific scenarios and taking into consideration respective lifestyles and needs should be part of communication strategies, and only followed by (compassionate) enforcement of many novel and changing civilian restrictions [78,79,80]. The individual “casualties” of COVID-19 related restrictions, often people already underprivileged [67], highlight the need to enhance efforts for empowerment and buy-in rather than relying heavily on enforcement, which may be susceptible to biases and unable to account for nuances in community needs and priorities. Finally, the best public health messages are the simplest ones that are feasible for everyone under different circumstances to adapt and adopt. Strategies for behavior change that have been proposed include [81,82]: (1) creating a mental model that captures the process of viral transmission to rationalise action; (2) creating social norms targeting self-identity and the influence of socially-modeled behavior; (3) creating the right level and type of emotion by complementing anxiety/disgust-triggering messaging with concrete advice on protective action; (4) replacing one behavior with another, so as to offer alternatives to risky yet often automatic actions e.g., “cough/sneeze into elbow/tissue vs. stop coughing/sneezing openly”; and (5) making the behavior easy, such as by pinning the new habit on an existing routine to minimise the learning curve. Finally, to ensure that public health messages are embraced, there is a need to both elevate understanding of infectious disease among the public, which COVID-19 has exposed to be inadequate and heterogeneous [83,84,85], as well as normalising uncertainty and modifications of scientific knowledge as part of public communication of science [83]. Such science literacy and scientific rationalization needs to be inculcated not only at a basic level through formal education but also ingrained through science communication and enculturation [86]. 

### 3.3. Increasing Acceptance and Combating Misinformation Surrounding New Tools Such as Vaccines

The COVID-19 pandemic not only caused unprecedented level of global disruption, it also spurred an unprecedented level of global and scientific cooperation to develop new tools, ranging from protective and monitoring devices, drugs, software applications, and vaccines. Despite the fact that most vaccines require over five years to develop from etiological discovery, the first COVID-19 vaccine took under 6 months to reach Phase III clinical trials and over three vaccines were approved for emergency use less than a year after the pandemic began [87,88]. What would have been a cause for celebration instead precipitated polarising views on vaccines and big pharma worldwide. Conspiracy theories and high-profile denouncements of the vaccines became widely shared on the internet and social media, resulting in growing hesitancy towards COVID-19 vaccines, even in countries with high disease burden [65,89,90].

Unlike communication about transmission dynamics of an unpredictable novel virus, communication regarding new tools can be better strategised even before the tools are ready for roll-out since their development is arguably within human or decision-maker control [91]. While there is no one strategy that can fully address vaccine hesitancy [92], early community engagement and social marketing to prepare for acceptance of tools such as COVID-19 vaccines [93] may cushion the pervasive misinformation or negative perceptions regarding vaccines, such as overemphasis of rare adverse events, persisting claims of debunked associations with autism, and concerns regarding ingredients deemed unnatural or not meeting religious requirements [94]. In particular, public concerns were apparent regarding mRNA vaccines, which were perceived as “untested” and developed without usual safety standards in place, despite the fact that mRNA technology has been used since the 1990s [95] before being harnessed as a novel vaccine platform to meet the urgent needs of the pandemic [96]. Some of the so-called safety arguments and outlandish conspiracy theories that target emotive reactions came from typical anti-vaccine sources, which could have been identified and managed before the misinformation became more widespread [97,98]. Others were less obvious, coming from segments of the scientific and academic community who claim not to be anti-vaccine, but stirred controversy with statements suggesting safety and efficacy standards were lowered in the quest for pharmaceutical companies to win a vaccine race [99]. These claims further exacerbated hesitance, even among undecided people [100], premised on the value judgment between facing the continued but unpredictable risk of becoming infected and falling ill with COVID-19 or being vaccinated with a vaccine that is perceived as “not 100% safe.”

The unfolding discourse of risk of vaccine side effects versus risk of COVID-19 underscores the need to re-evaluate the basic understanding and the emotional response to scientific concepts for the majority of the population [101], and address these fundamental gaps. Too often the language of science which can only speak in terms of probability and uncertainty is lost on people who want to be reassured in totality, while demanding transparency. This is where risk communication strategies, such as those framed by the psychometric paradigm, are key to bridging the risk perception gap between experts and non-experts [102]. Without understanding of real-world limitations in ethical data acquisition and appreciation that decisions are made based on a balance of scientific and administrative/practical risk-benefit trade-offs, even the most transparent reports (as was required for many of the COVID-19 manufacturers) would do little to assuage public concern. However, there are indications that influencers and social media savvy experts communicating simplified messages using personal and creative means to demystify the science and processes behind mRNA and other COVID-19 vaccines, and reiterate key messages of vaccine effectiveness and safety while calling for societal uptake and solidarity to end the pandemic may resonate better with the public [80,103,104]. Indeed, the captive interest in vaccines in the context of COVID-19 presented an opportunity to introduce concepts such as efficacy and herd immunity into the larger public psyche. Nevertheless, such efforts to address misinformation are not devoid of challenges including the speed of changing trends, limited resources, and difficulty in measuring actual reach [105]. Besides more targeted strategies to curb social media perversion from strong anti-vaccine networks [106,107], the rampant spread of misinformation amongst vaccine undecided individuals raises the question to what extent public susceptibility to vaccine skepticism reflects general inadequacies in health and science literacy (including basic understanding of immunology) or public mistrust [86,89,107]. In either case, better communication and engagement may help address future battles against misinformation-driven hesitancy towards new biomedical tools [108].

## 4. Sustaining Public Health Interventions towards Pandemic Recovery

### 4.1. Active Management of Social Stigma and Mental Health Repercussions

The emergence of COVID-19, a new and highly contagious disease, stimulated high anxiety and fear. Yet, it was also soon established that infection was only potentially severe in certain groups in the population, with a larger majority manifesting mild or no symptoms. This combination of indiscriminate risk of infection but imbalance in risk of disease quickly brewed some negative social sentiments ranging from apathy, polarisation, frustration, judgment, and worse, discrimination against people seen to pose risk of infection. COVID-19 patients in general faced varying levels of social stigma, exacerbated further by any existing race-, religion- or social class-based discrimination [109]. Furthermore, healthcare professionals in general were at particular hazard of stigma and bullying [110,111], being seen at the forefront of the COVID-19 pandemic and consequently, a likely victim and source of new infections. This was worsened by media reports of “super spreaders”, a term that has been found to be particularly vague and problematic [112], implying blame on individuals and indirectly inciting public animosity and even harassment. Such stigmatising attitudes had grave consequences on transmission control, as people then had the choice of getting tested and possibly confirmed, isolated, and stigmatised as COVID-19 patients, or ignoring their exposure and/or symptoms while spreading the virus if they were indeed carrying an undiagnosed infection. This highlights the need to incorporate strategies that address stigma upfront in infectious disease control through inclusive policies and empathetic communication [113], especially when conducting COVID-19 screenings at public points of entry, as part of COVID-19 recovery and re-integration into a post-pandemic world. Leaders from all levels of governance, organizations and groups play a role in establishing healthy spaces devoid of discriminatory elements. The tone and language in the media and by social media influencers that reduce stigmatising labels, and promote empathy in intrapersonal and interpersonal interactions for all, play crucial mitigatory roles [112]. Providing space for COVID-19 survivors to anonymously or openly share their experiences and perspectives may also be a strategy to de-stigmatise and curb this irrational fear of the unknown or “other” [109]. 

Beyond stigma, the pandemic has taken an immeasurable toll on mental health and wellbeing of individuals across the globe. The highly charged atmosphere surrounding COVID-19—characterised by widespread signals of alarm on public media and in social circles, disruptions to daily life activities (work, school, social interactions), economic uncertainty, and bottlenecks in healthcare—extends the burden of the disease beyond those who contracted it. Manifestations of mental health decline include anxiety, insomnia, perception of insecurity, anger, increased use of health devices for fear of illness, and behaviors risky for health (increased consumption of alcohol, illicit drugs and tobacco, change in work-life balance, social isolation, increased family conflicts and violent behavior). Healthcare workers also reported the psychological burden of making difficult triage decisions when resources were stretched and also fearing for their own safety. The psychological impact of COVID-19 on the general population parallels those previously seen during SARS, natural disasters, and even armed conflicts [114], hence the road to recovery will require systematic mitigation of psychological risk factors surrounding the pandemic. To sustainably address the perception of isolation, helplessness, loss of normalcy, and collective panic, strategic communication needs to be deployed to elicit a thoughtful crisis response with strong emphasis on the steps toward recovery and consistent highlights of ordeals overcome, rather than a continuous feed of daily case numbers, death tolls, and footage of empty store shelves [115]. Although quality of life and access to healthcare (particularly psychiatric care) have generally declined, communication channels that remain open must be leveraged to deliver means of support across communities, whether in the form of grounded, actionable advice, delivery of care/medication, or opportunities to establish productive social connections with others despite the restrictions of stay-at-home orders. 

### 4.2. Maintaining Resilience, Self-Efficacy and Solidarity in the Face of Pandemic Fatigue and Emerging Concerns

Lockdowns and strict restrictions were always known to be temporary solutions to controlling COVID-19 [61,66]. However, over a year into the pandemic, several cities that enjoyed very low transmission and relaxed restrictions have gone in and out of various forms of lockdown in response to changes in COVID-19 incidence. While some governments applied enhanced restrictions more urgently than others (often to their success) [116], others have relegated to half-imposed, drawn out restrictions that seem less effective. Regardless of lockdown impositions, several interventions require long-term commitment even after community transmissions are significantly controlled, until the virus is locally eliminated. These include social and physical distancing measures, interstate and international travel restrictions, hygiene and mask-wearing. Where communication of early “heroic” responses to an arising emergency appeared more focused and unified [117], in later stages messages appear to have become diluted and simultaneously, commitment to adherence significantly waned even in settings where COVID-19 continues to spread rampantly. This is arguably the manifestation of pandemic fatigue stemming from reduced perception of risk and/or perception of benefit [117,118] and any threats to communal solidarity, which along with premature policy decisions regarding border openings and mass gatherings, has been implicated in subsequent waves of COVID-19 around the world [119]. 

As case numbers continue to decline due to successful implementation of public health interventions and tools, risk perception may decline correspondingly. This then challenges any need to quickly respond to re-emergent threats such as local outbreaks that require drastic implementation of measures to be rapidly contained. Indeed, sustaining pandemic control poses unique challenges due to labor and expense required to maintain surveillance and vaccination coverage against a perceived diminishing threat, as seen in the case with measles [120]. The perception that the pandemic is under control or that the disease has disappeared, coupled with a dip in confidence from an intervention like a vaccine with new and yet explained severe side effects, could quickly jeopardise the required coverage to maintain a herd effect and potentially re-catapult the country into a new epidemic [121]. This has occurred alarmingly in India, but also Taiwan, Singapore, Thailand, Malaysia and other countries which previously achieved COVID-19 control, but then faced sudden spikes in light of loosened restrictions, introduction of highly transmissible variants, amidst a lagging vaccine roll-out [122,123]. However, this is where continuous communication strategies and community engagement play a particularly important role in addressing arising concerns that may lead to resistance against vaccines and other public health tools, whilst preparing the public to remain resilient and adaptable to temporary albeit strict measures to avoid resurgence [115,124]. Australian states, all of which have largely eliminated local transmission of COVID-19 by the end of 2020 through strict border control and mandatory quarantine accommodation [13,125], have shown success in applying snap lockdowns at early signs of COVID-19 transmission into the community. Combining contact tracing, free mass testing, universal healthcare coverage, and economic relief with public and community engagement, including government-supported Indigenous Australian-run COVID-19 response center [126,127], efforts that ask for temporary community sacrifices to protect the post-pandemic way of life that they have worked hard to regain and uniquely enjoy [128], have thus far been able to swiftly contain new risks of resurgence. 

Nevertheless, Australia, like many countries, continue to face significant challenges in delivering COVID-19 vaccines. This has been due to a combination of supply and administration issues [129], but early in 2021, reports of an extremely rare but deadly thrombosis with thrombocytopenia syndrome (TTS) linked to vaccination using the AstraZeneca vaccine among younger adults, further complicated roll-out due to reduced public confidence in this vaccine [130,131]. Even in countries where the risk of COVID-19 remains high [132], hesitancy towards the viral-vectored vaccine caused many to now opt to wait for mRNA vaccines. The early decision by European countries to suspend the use of the viral vectored vaccines, followed by different countries applying varying age group recommendations and/or dropping viral-vectored vaccines from the national program altogether [133], and crucially, poor communication surrounding this potential side effect indubitably fueled the impression that the viral-vectored vaccines were less safe than their counterparts. 

As the vaccines have only been shown protective against disease and not infection, elimination strategies through herd immunity will likely require extremely high proportions of vaccinated populations [134], without which resurgence remains a risk as seen with the increased incidence in Seychelles after re-opening borders despite having 63% of the population fully vaccinated [135,136]. Albeit the rare TTS is now better managed and messages of vaccine safety are being reinforced [131,137], the negative impact of initial poor vaccine safety messages may be difficult to reverse and the cost of the communication failure early in the vaccination roll-out remains to be seen as the COVID-19 pandemic continues to ravage many countries across the globe in 2021 [138].

## 5. Further Lessons and Questions

COVID-19 emerged as a potential pandemic late December 2019. With the exception of close neighbors to China such as Taiwan and Hong Kong, as well as New Zealand, strict border control measures were only put in place several months later elsewhere across the globe [61,66]—likely due to underestimation of the potential magnitude of the problem. The reluctance against closing borders and instating lockdowns are typically argued from the socioeconomic perspective, but as COVID-19 (and previously the flu pandemic) has shown [139], early action delivered with effective communication to contain the spread of the virus would ultimately lead to faster epidemic and socioeconomic recovery [140,141,142].

Beyond border closures, other aspects of Taiwan’s early response were exemplary in preventing community transmission or the need for a lockdown thus far. On top of ramping up supplies, diagnostic testing, and enforcing mandatory mask-wearing, Taiwan’s COVID-19 dedicated Central Epidemic Command Center (CECC) established since late January 2020 also made fighting misinformation a key part of the CECC’s mandate, holding tight control as the only source for COVID-19 related information in Taiwan to prevent the spread of fake news [56]. Standout aspects of Taiwan’s CECC approach is the use of data analytics to identify trends in misinformation, “humor over rumor” campaign strategies [143], and a dedicated toll-free number for the public to call in with any queries or suggestions related to COVID-19, which became a means for crowd-sourcing ideas whilst also obtaining popular support [56,144]. Notwithstanding the advantages of Taiwan being a smaller, largely digitally-connected and homogeneous social democracy, it begs the question whether such centralised health communication strategies would also be effective in countries with different forms of government and larger more heterogeneous demographics.

Traditionally, health communication frameworks are mainly premised on the health belief model which targets behavior change at the individual level e.g., patient or caregiver [145,146,147]. When it comes to eliciting rapid widespread behavior change at a societal level, such communication strategies fall short of meeting the more complex and challenging needs of communities. The greater number of stakeholders with diverse profiles and social roles contribute to this complexity, requiring a much more consolidated effort from multiple active agents [148]. Given there was no single agency with position of authority over the subject of COVID-19, it fell upon various experts in virology, infectious diseases, biomedical research, computational science, environmental sciences, engineering, social science, economics, and governance to contribute their critical pieces of the global puzzle [149]. Their diverse expertise had to be rapidly consolidated to provide guidance on specific aspects of the problem which then had to be communicated to provide the public grounded information and clear directions amidst a concurrent and confounding infodemic [10,150]. While some countries were able to leverage different entities towards a synergistic solution (notably where various field experts worked in partnership with their governments), many others struggled to achieve this form of consolidation among the disparate silos of knowledge and priorities, putting them at a disadvantage when it comes to comprehensively assessing the risks of and protective responses to COVID-19 in the community. As a result, these societies were at the mercy of knee-jerk reactions to the crisis in the form of poorly-structured lockdowns and movement restrictions that are further complicated by a widespread trust deficit towards political authority and mainstream information channels [64,65]. Importantly, better communication and coordination between different agencies and expert groups should precede health education for the public, so as to avoid confusion in policies or worse, conflicting procedural policies that ultimately defeat their intended purposes to control disease and protect public health [151].

Another key theme in combating COVID-19 is the importance of trust, empathy and compassion in crisis communication at various levels [152,153]. With regard to leadership, communities were more likely to embrace and trust decisions communicated with compassion. Prime Minister Jacinda Ardern’s compassionate data-grounded leadership was highlighted to be a key factor in New Zealand’s success in early control of the pandemic [154]. Consistent positive communication from leaders, backed by health and science experts across different countries, not only was used to keep everyone informed, but also became a platform for establishing rapport, social solidarity and public trust [154,155]. By systematically addressing the social machinery of the public, driven by strong needs for identity, values, reciprocity, trust, dignity, and emotional security, communicating health during a pandemic can rise above its current reputation as a top-down dictum with seemingly alien, counterintuitive, and restrictive elements [115]. This can only be achieved by upholding the principles governing successful behavior change advocacy such as supporting the public in conceptualising the risk elements and purpose of change, engineering room for protective social norms to leverage any positive ripple effects, striking a balance between anxiety and agency through appeals to emotion in urgent yet uplifting messaging, offering “freedom within boundaries” to guide the public in making appropriate choices, and scaffolding the habit-building process with constructive support towards self-efficacy [81,82].

With the mysterious COVID-19 unfurling in the era of social media, the nuanced impact of communications on social networks and multilateral information sharing on the success and challenges in various efforts for COVID-19 control cannot be trivialised. Whether the influence of social media communications was a boon or a bane for epidemic control remains debatable [156,157]. There were clear instances where social media (and influencers) played a positive role in vulgarisation of infectious disease and public health concepts such as the viral reproductive number (R0) and the need to “flatten the curve” [158], which likely enabled better acceptance of public health interventions. Conversely, the speed and frequency at which various types of (mis)information were shared at times became a source of communication fatigue and overload, not to say the least of resurfaced debunked conspiracy theories or reinvigorated misunderstanding. It remains unclear to what extent many of the often cyclical conversations on issues intersecting COVID-19 control and social interest, such as debates on whether available COVID-19 vaccines prevent infection or disease or superiority of one vaccine over others, reflect empowering discussions or exhausting distractions. Additionally, although predominant use of the internet and social media for various purposes has been useful in sustaining communications and connectivity whilst reducing transmission risks arising from physical contact, this may be at the expense of continued neglect of community members who are not digitally connected and unable to access online spaces and resources [159].

Finally, to truly prevent the emergence and spread of viral diseases such as COVID-19, the delicate web of human action and its impact on environmental, biosphere integrity, and the health of the people and the planet must continue to be communicated and imprinted into our social psyche. This includes targeted awareness campaigns on aspects of conservation and sustainability, infectious disease and One Health for different age and stakeholder groups. Further, the elephant in the room, the barriers in communication between disciplines, stakeholders and stewards, must be actively addressed. As long as problems are only approached unilaterally, we may be doomed to constantly miss the forest for the trees, and find ourselves painfully underprepared even for the next pandemic. While some aspects of silos may be necessary due to the practicality of compartmentalization in pursuit of science, its restrictive effects arise from lack of opportunity to interact and transfer knowledge [160]. A positive theme arising from COVID-19 response worldwide was the opportunity for traditionally separate disciplines and groups to work together to solve a common problem. Thus, to promote transdisciplinarity in our problem solving, cross-pollination through thematic rather than discipline-based platforms and initiatives may be one approach. More importantly, efforts to perhaps establish a “minimum common technical translation” across disciplines while equipping experts with strategies in science communication to converse with stakeholders and stewards may better facilitate transdisciplinary discussions and public engagement in the future. Our silos and disconnects are not the product of scientific failure, they are manifestations of communication breakdown. Only when we alleviate underlying barriers and enhance enablers to fulfill the need for effective communication at different phases (Figure 1), will we stand a chance to reach One Health goals and prevent future pandemics.

## 6. Conclusions

Clearly, in the case of COVID-19, the stellar growth in scientific knowledge and progress in development of tools to mitigate the spread and impact of the disease is only half of the battle won. An equally important challenge to overcome is ensuring that communication of the knowledge and tools acquired are vulgarised in a way that enables public health strategies to succeed with not only regulation and enforcement, but through community engagement and support. While it bears to say we must communicate more effectively, the methods for communication about viral diseases and their means of control are also scientific challenges, which require better consolidated research to determine effective strategies for bridging different disciplines and groups of people under different contexts. Ultimately, the aim of effective communication is not to overwhelm people with information that they have difficulty making sense of, but instead to elevate their self-efficacy to act on the information they have. Moving forward, building trust and harnessing transdisciplinary voices that deliver clear, empathetic, and actionable messages using effective communication tailored for different purposes and audiences is critical for prevention and control of future viral diseases, particularly those that are as explosive as COVID-19.

## Figures and Tables

**Figure 1 viruses-13-01058-f001:**
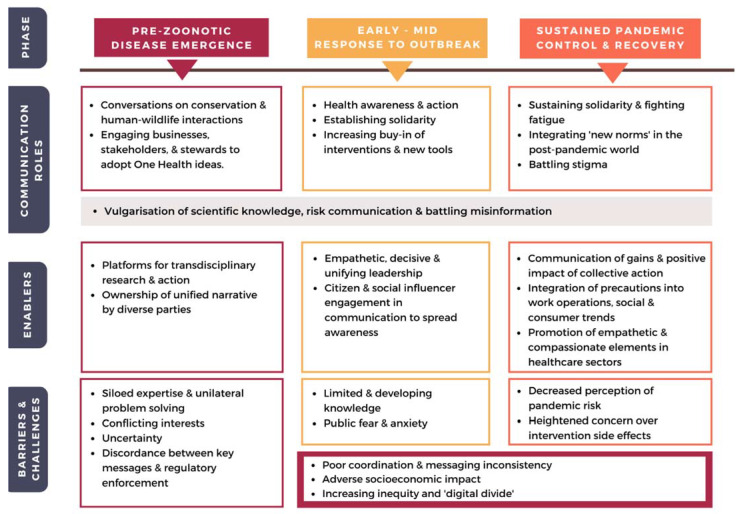
Summary of communication roles and their enablers and barriers at different phases of the COVID-19 pandemic.

## Data Availability

Not applicable.

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
