# Peer review of "Effective Communication at Different Phases of COVID-19 Prevention: Roles, Enablers and Barriers"

_viruses, 2021, doi:10.3390/v13061058_

Round 1

Reviewer 1 Report

The authors brought a wealth of insights on the evolution of the COVID-19 pandemic and discusses implications/considerations for health/risk communication.  They correctly urge international decision makers to place communication front and center in pandemic control efforts and offered suggestions on strategies for more effective communication.   I have the following comments which I hope will strengthen the paper: 1) I would discuss the changes in communication and behaviors over the last 18 months more, including the possibility of "pandemic fatigue", evolving landscape of vaccine distribution and vaccine hesitancy, and mental health implications that communication efforts should take into account; 2) as authors observed, different countries and jurisdictions enacted vastly different approaches to behavioral mitigation strategies and communications about them, essentially creating a global natural experiment.  I would encourage the authors not only to describe the early 2020 measure (e.g. successes in Taiwan and New Zealand and mixed signals in Western countries) but to integrate the vaccine availability and policies in late 2020 and early 2021 into the discussion. 

Author Response

Reviewer 1

1) The authors brought a wealth of insights on the evolution of the COVID-19 pandemic and discusses implications/considerations for health/risk communication.  They correctly urge international decision makers to place communication front and center in pandemic control efforts and offered suggestions on strategies for more effective communication.  

We appreciate the reviewer’s positive response towards our attempt to frame the paramount importance of communication in pandemic control, as well as potential strategies that we have gleaned from the literature and our own experience being part of this unfolding COVID-19 pandemic.

2) I have the following comments which I hope will strengthen the paper: 1) I would discuss the changes in communication and behaviors over the last 18 months more, including the possibility of "pandemic fatigue", evolving landscape of vaccine distribution and vaccine hesitancy, and mental health implications that communication efforts should take into account;

We thank the reviewer for this valuable suggestion. We have included additional discussion on the role of communication in pandemic fatigue, and expanded further on developments regarding to vaccine roll-out and public confidence (especially in light of issues such as the extremely rare blood clotting side effects from Astra Zeneca/adenovirus vectored vaccines), and communications addressing pandemic repercussions on mental health in the revised manuscript.

3) as authors observed, different countries and jurisdictions enacted vastly different approaches to behavioral mitigation strategies and communications about them, essentially creating a global natural experiment.  I would encourage the authors not only to describe the early 2020 measure (e.g. successes in Taiwan and New Zealand and mixed signals in Western countries) but to integrate the vaccine availability and policies in late 2020 and early 2021 into the discussion.

We thank the reviewer for this suggestion. The differences in roll-out and accessibility of vaccines is certainly a key factor in pandemic development late 2020- early 2021, and we have attempted to also include comments on the role of communication in the vaccine roll-out, primarily related to building trust and confidence in the effectiveness and safety of vaccines, in the revised manuscript.

Reviewer 2 Report

This manuscript is very interesting to understand the pandemic crisis caused by COVID-19.

I think the authors have done fine research. I suggest improving the discussion about emotions and citizens' policy using more accurate references from social sciences.

I recommend revising these papers to enrich your text:

Belli, S., & Broncano, F. (2017). Narratives of trust: sharing knowledge as a second-order emotion. Human Affairs27(3), 241-251.

Belli, S., & Raventós, C. L. (2021). Collective subjects and political mobilization in the public space: Towards a multitude capable of generating transformative practices. Human Affairs31(1), 59-72.

Elcheroth, G., & Drury, J. (2020). Collective resilience in times of crisis: Lessons from the literature for socially effective responses to the pandemic. British Journal of Social Psychology59(3), 703-713.

Li, C., Chen, L. J., Chen, X., Zhang, M., Pang, C. P., & Chen, H. (2020). Retrospective analysis of the possibility of predicting the COVID-19 outbreak from Internet searches and social media data, China, 2020. Eurosurveillance25(10), 2000199.

Romano, A., Spadaro, G., Balliet, D., Joireman, J., Van Lissa, C., Jin, S., ... & Leander, N. P. (2021). Cooperation and trust across societies during the COVID-19 pandemic. Journal of Cross-Cultural Psychology, 0022022120988913.

Author Response

Response to Reviewer 2:

1)This manuscript is very interesting to understand the pandemic crisis caused by COVID-19.

We thank the reviewer for this positive response to our manuscript.

2) I think the authors have done fine research. I suggest improving the discussion about emotions and citizens' policy using more accurate references from social sciences.

I recommend revising these papers to enrich your text:

Belli, S., & Broncano, F. (2017). Narratives of trust: sharing knowledge as a second-order emotion. Human Affairs27(3), 241-251.

Belli, S., & Raventós, C. L. (2021). Collective subjects and political mobilization in the public space: Towards a multitude capable of generating transformative practices. Human Affairs31(1), 59-72.

Elcheroth, G., & Drury, J. (2020). Collective resilience in times of crisis: Lessons from the literature for socially effective responses to the pandemic. British Journal of Social Psychology59(3), 703-713.

Li, C., Chen, L. J., Chen, X., Zhang, M., Pang, C. P., & Chen, H. (2020). Retrospective analysis of the possibility of predicting the COVID-19 outbreak from Internet searches and social media data, China, 2020. Eurosurveillance25(10), 2000199.

Romano, A., Spadaro, G., Balliet, D., Joireman, J., Van Lissa, C., Jin, S., ... & Leander, N. P. (2021). Cooperation and trust across societies during the COVID-19 pandemic. Journal of Cross-Cultural Psychology, 0022022120988913.

We sincerely appreciate the specific suggestions by the reviewer to improve references, and have added these in the manuscript related to the social science aspect of pandemic control. Additionally, we have further edited the revised manuscript for length (where possible) and clarity.